# Hospital Production of Sterile 2% Propofol Nanoemulsion: Proof of Concept

**DOI:** 10.3390/pharmaceutics15030905

**Published:** 2023-03-10

**Authors:** Amélie Cèbe, Bérangère Dessane, Pauline Gohier, Jean-Marc Bernadou, Arnaud Venet, Fabien Xuereb, Sylvie Crauste-Manciet

**Affiliations:** 1Pharmaceutical Technology Department, Bordeaux University Hospital, F-33600 Pessac, France; 2INSERM, Biologie des Maladies Cardiovasculaires, University Bordeaux, U1034, F-33600 Pessac, France; 3Univ Angers, CHU Angers, INSERM, CNRS, MINT, SFR ICAT, UMR INSERM, F-49000 Angers, France

**Keywords:** drug shortages, propofol, nanoemulsion, sterile, hospital preparation

## Abstract

In the context of essential drug shortages, this article reports a proof of concept for the hospital preparation of a 2% propofol injectable nanoemulsion. Two processes for propofol were assessed: mixing propofol with the commercial Intralipid^®^ 20% emulsion and a “de novo” process performed using separate raw materials (i.e., oil, water, and surfactant) and optimized for droplet size reduction with a high-pressure homogenizer. A propofol HPLC-UV stability-indicating method was developed for process validation and short-term stability. In addition, free propofol in the aqueous phase was quantified by dialysis. To envision routine production, sterility and endotoxin tests were validated. Only the “de novo” process using high-pressure homogenization gave satisfactory physical results similar to commercialized Diprivan^®^ 2%. Both terminal heat sterilization processes (121 °C, 15 min and 0.22 µm filtration) were validated, but an additional pH adjustment was required prior to heat sterilization. The propofol nanoemulsion was monodisperse with a 160 nm mean droplet size, and no droplets were larger than 5µm. We confirmed that free propofol in the aqueous phase of the emulsion was similar to Diprivan 2%, and the chemical stability of propofol was validated. In conclusion, the proof of concept for the in-house 2% propofol nanoemulsion preparation was successfully demonstrated, opening the field for the possible production of the nanoemulsion in hospital pharmacies.

## 1. Introduction

In the context of drug shortages, the feasibility study of the hospital pharmacy production of drugs formulated as colloidal dispersions, such as nanoemulsions (NEs), is of great interest. NEs have been widely used for parenteral nutrition for the last 40 years and, more recently, as a vehicle for drug delivery. Among the few drugs formulated as NEs, propofol has retained all of our attention, which has been especially highlighted by the shortage of essential anesthetic drugs during the COVID-19 pandemic [1], i.e., curare, midazolam and propofol. If the production of injectable aqueous solutions of curare were likely to be transposable to hospital processes, the production of injectable nanoemulsions (NEs) is considered a major challenge, given the high level of skills and technology required to produce sterile nanodroplet emulsions.

Injectable NEs are colloidal dispersions of oil droplets dispersed in an aqueous phase as oil-in-water (o/w) NEs stabilized by surfactants. In contrast to injectable solutions, microemulsion NEs are thermodynamically unstable systems [2,3], and major attention must be paid to their physical stability (e.g., aggregation, creaming and droplet growth). However, compared to emulsions, NEs are recognized to be kinetically stable systems [3]. Well-known injectable NEs (e.g., Intralipid^®^) are currently administered for parenteral nutrition [4]. The stability of colloidal dispersions is known to be affected by their electronic environment, and the incorporation of a drug may impact their physical stability [5].

Nevertheless, considering our strong knowledge of parenteral emulsion design and stability [6,7,8], we decided to work on the feasibility of the process for hospital emergency preparation.

Different possibilities of production were envisioned [9]: the easiest, in the hospital context, was the addition of a propofol oil solution to a commercialized NE (extemporaneous addition method), and the second method was the “de novo” method, where the ingredients are incorporated into oil and aqueous phases prior to the emulsification process and droplet size reduction. To obtain a sterile emulsion, heat sterilization was considered the best method, but we also assessed the possibility of using an aseptic process with 0.22 µm sterile filtration. For each method, we assessed the physical characteristics of oil droplets, i.e., the granulometric properties, zeta potential and stability of the emulsion. In addition, we developed a stability-indicating method for propofol to measure the impact of both the preparation process and storage on the chemical stability of the drug. Finally, for the potential tolerance issue [10,11], we assessed the residual free propofol concentration in the aqueous phase.

Here, our study paves a path for the contribution of hospital pharmacies in an agile network to face potential future shortages of essential drugs designed as nanoemulsions.

## 2. Materials and Methods

Propofol (2,6 di-isopropylphenol) for tests was purchased from Sigma-Aldrich (St. Louis, MO, USA), the Propofol pharmaceutical raw material was purchased from Pharma Chemicals SA (Parc industriel 19, 1440 Wauthier-Braine, Belgium), glycerol was purchased from Coopération pharmaceutique française (Melun, France), and soybean oil was kindly provided by ADM Nutrition (Health & Wellness ADM-SIO02, Rue George Sand—Building A Fourqueux 78112, France). Lipoid E80 SN was kindly provided by Lipoid GmbH (Ludwigshafen, Germany).

### 2.1. Propofol Nanoemulsion Production Processes

#### 2.1.1. Extemporaneous Addition Method

A propofol oil solution was extemporaneously added at 25 °C to the commercial Intralipid^®^ NE and mixed using a high-shear mixer (Ultra-Turrax^®^; IKA^®^, Staufen, Germany) for 20 min.

#### 2.1.2. “De Novo” Method

Egg lecithin was dispersed in the oil phase, which was a mixture of soybean oil and propofol. The oil and water phases were heated separately and brought to the same temperature in a water bath at 70 °C until the complete dissolution of lecithin. Emulsification was performed by phase inversion by adding the water phase to the oil phase using a high-speed mixer (Ultra-Turrax^®^; IKA^®^, Staufen, Germany) for 20 min. The coarse emulsion was then obtained, and the reduction in the droplet size was obtained by using a high-pressure homogenizer (Lv1 Microfluidizer^®^; Microfluidics™, MA, USA) set at 1500 bar (22,000 psi) to obtain a stable monodisperse NE with droplets smaller than 200 nm. pH was adjusted in the range 6–10 by adding 0.05 N sodium hydroxide, and osmolality was adjusted by adding glycerol. All samples subjected to heat sterilization were distributed in type 1 transparent glass vials (Schott adaptiQ^®^, Adelphi, Milan, Italy) closed with rubber stoppers (West FluroTec^®^, Adelphi, Milan, Italy) and aluminum crimp seals (Flip-Off^®^ CCS, Adelphi, Milan, Italy). The minimum number of passes through the microfluidizer^®^ was determined by measuring the mean droplet size (mds) and polydispersity index (pdi) using a dynamic light scattering Zetasizer^®^ Nano ZS (Malvern Instruments, Malvern, UK). Large droplets above 1 µm and 5 µm were additionally researched using laser diffraction (Mastersizer^®^, Malvern, UK).

#### 2.1.3. Sterilization Methods

Sterilization of the propofol NE was performed with a final heat sterilization using a heat sterilizer (PROHS^®^ EV-150 SA, Maia, Portugal) at 121 °C for 15 min in accordance with the European Pharmacopeia. pH adjustment of the water phase of the NE was optimized by comparing the results of the pH and granulometric parameters (mds, pdi) of the NE before and after heat sterilization.

Alternatively, the feasibility of sterile filtration though a 0.22 µm filter (Sterivex^®^ GP Millipore, Merck KGaA, Darmstadt, Germany) was also assessed.

For both methods, the final propofol concentration was determined before and after the sterilization process.

#### 2.1.4. Propofol Nanoemulsion Compositions

The objective was to produce a formulation as close as possible to the commercial propofol NE. We considered the Diprivan^®^ formulation as the reference. Table 1 provides the formulations performed according to the process used, i.e., the “extemporaneous addition” or “de novo” method.

#### 2.1.5. Physico-Chemical Characterizations

Visual observation. NEs were visually inspected to detect any oiling out and phase separation.Granulometric analysis and zeta potential. The hydrodynamic size was determined using a DLS device from Malvern Instruments (Zetasizer^®^ Nano ZS) in the NE diluted at 1:1500 (*v*/*v*). The mean hydrodynamic size was determined from 3 independent measurements performed at 25 °C. Additionally, to assess the lack of droplets above 1µm and 5µm, granulometric analysis was performed using a laser diffraction particle size analyzer (Mastersizer^®^; Malvern Instruments Ltd., Malvern, UK). Each sample was diluted in water to an appropriate concentration before measurement at 25 °C. Zeta potential measurement was performed on a 1:1500 (*v*/*v*) diluted NE sample using a Zetasizer Nano ZS coupled with a Folded Capillary Cell (DTS1060) from Malvern Instruments.pH and osmolality. The pH was measured by potentiometry (SevenCompact™ S210, Mettler Toledo, OH, USA). Osmolality was determined by cryoscopy (Löser™ type 15 osmometer, Löser Messtechnik, Berlin, Germany). Measures were repeated 3 times.HPLC assay. Propofol quantification was developed according to the following HPLC method. HPLC-UV UltiMate™ 3000 (Thermo Scientific^™^) with a DAD-3000 detector and a Phenomenex^®^ Kinetex^®^ (F5: 2.6 µm 100 Å 100 × 4.6 mm) column. The mobile phase was acetonitrile–methanol–water (13:54:33, *v*/*v*/*v*) delivered at a flow rate of 0.8 mL/min; the ultraviolet-light absorbance detector was set at 273 nm. The temperature of the column was set at 40 °C, and the run time was 13 min. The temperature of the autosampler was fixed at 10 °C to prevent the evaporation of the samples (methanol solvent). Prior to HPLC analysis, the NE was diluted 1/1000 in methanol, resulting in the complete destruction of the NE. The method was validated according to ICH guidelines Q2(R1) [12] by determining its linearity, accuracy and precision. Linearity was established using 5 calibration solutions of 12, 16, 20, 24 and 30 µg/mL, corresponding to 60–155 % of the working solution. Linear regression was performed using data analysis software. Linearity was determined using the correlation coefficient (R^2^). Precision was determined using the relative standard deviation or coefficient of variation (CV%), as defined by ICH. Precision (CV%) and accuracy (%) were determined using three quality controls (14, 22, 28 µg/mL) prepared with another stock solution. Intra-day repeatability (precision and accuracy) was determined using the three quality control solutions (in methanol) repeated six times per day. Inter-day repeatability (precision and accuracy) was determined by repeating the experiments on three days. Peak match values were determined using Thermo Scientific^TM^ Dionex^TM^ Chromeleon^TM^ 7 Chromatography Data System version 7.2.7.Free propofol determination. Propofol NEs were dialyzed on Rapid Equilibrium Dialysis (RED) Device Inserts (89809, Thermo Scientific™, Rockford, IL, USA), with a cut-off MW of 8000 daltons.

To balance the tonicity with the NE in the receptor medium, a solution of glycerol 2.5% (*w*/*v*) in water was used as receptor medium (Glycerol, Cooper, Melun, France).

The sample volume in the sample compartment (propofol) was 300 µL, and the receptor medium volume (glycerol 2.5%) in the dialysate compartment was 550 µL. Dialysis was performed at 25 °C for 420 min with agitation at 400 rpm.

The receptor medium was analyzed by high-performance liquid chromatography (HPLC) after a 1:5 (*v*/*v*) dilution in methanol. The determination of total propofol and free propofol in the NE was performed by the HPLC method described above. The time-dependent partitioning of propofol between the propofol NE and the receptor solution was validated on the Diprivan^®^ 2% commercial emulsion. The kinetics of the dialysis showed the stabilization (plateau) of free propofol concentrations between 330 and 480 min in the receptor compartment (Table 2). For practical reasons, the duration retained for further dialysis of the samples was 420 min (plateau).

#### 2.1.6. Stability Studies

NE physical stability. Granulometric analysis was conducted on NE for 1 month. Additionally, to detect droplets larger than 1µm, NEs were analyzed by laser diffraction using a Mastersizer^®^ 3000 particle size analyzer (Malvern Instruments, Malvern, UK). Values quoted are the average of 3 measurements +/− standard deviation (SD).Propofol chemical stability. Propofol quantification and research on the degradation products were performed using the HPLC method described previously. The stability-indicating nature of the HPLC analytical method was validated following the ICH and SFPC-GERPAC guidelines [13].

Forced degradation study. A forced degradation study of propofol was performed to obtain a quantitative and qualitative stability-indicating analytical method. Tests were performed under four stress conditions (acidic, alkaline, oxidative, and photolytic) on a 0.02 mg/mL solution of propofol in methanol (Propofol Pharmaceutical, Pharma Chemicals). This solution was obtained by diluting 20 mg of propofol in QSP 100 mL of methanol. Degraded samples were compared with a sample of the same undegraded solution after analysis by HPLC-UV.

Acid degradation conditions. The acid degradation of propofol was tested using a hydrochloric acid (HCl) solution by adding 1000 µL of a 1 M aqueous HCl solution to 1000 µL of a 0.02 mg/mL propofol solution in methanol. The resulting propofol-in-HCl mixture was heated for 60 min at 80 °C and then neutralized with 1000 µL of 1 M sodium hydroxide (NaOH). Then, 1000 µL of methanol was added for the analysis of the degraded propofol solution by HPLC-UV. Alkaline degradation conditions. The alkaline degradation of propofol was tested using a sodium hydroxide (NaOH) solution by adding 1000 µL of a 1 M aqueous NaOH solution to 1000 µL of a 0.02 mg/mL propofol solution in methanol. The resulting propofol-in-NaOH mixture was heated for 60 min at 80 °C and then neutralized with 1000 µL of 1 M hydrochloric acid (HCl). Then, 1000 µL of methanol was added for the analysis of the degraded propofol solution by HPLC-UV. Oxidative degradation conditions. The oxidative degradation of propofol was tested using a 30% hydrogen peroxide (H_2_O_2_) solution (110 volumes). The resulting 0.02 mg/mL propofol-in-H_2_O_2_ mixture was heated for 120 min at 80 °C. Then, 2000 µL of methanol was added for the analysis of the degraded propofol solution by HPLC-UV analysis. Light degradation conditions. For forced light degradation, the 0.02 mg/mL propofol solution was exposed to a fluorescent light (29,000 lux) placed at a distance of 15 cm from the sample for 220 h. The analysis of the degraded propofol solution was performed by HPLC-UV.

Short-term stability of propofol. The chemical stability of propofol was determined for 1 month of storage at 25 °C, protected from light. The stability study was conducted in triplicate over three different batches, as recommended by ICH Q1A(R2) [14]. The area under the curve (AUC) at each time point was measured, and variations in concentration were determined. Stability was defined as the concentration remaining stable between 90 and 110% of the initial concentration over the 30 days of storage, according to ICH [15], and degradation products were researched and quantified (%) in relation to the propofol area under the curve.

#### 2.1.7. Sterility Test

A sterility test was performed according to European Pharmacopeia chapter 2.6.1 (EP) [16] in triplicate at D0 after heat sterilization using the filtration method (Steritest^™^ Merck). Trypticase soy broth was incubated for 14 days at 37 °C (±2 °C) and 25 °C (±2 °C) and observed daily to detect microbiological growth. The sterility test was passed when no growth was observed after 14 days of incubation.

Prior to the sterility test, a media growth promotion test was performed with the six microorganisms recommended by the European Pharmacopeia: *Staphylococcus aureus* (ATC 6538); *Bacillus subtilis* (CCM 1999); *Pseudomonas aeruginosa* (ATC 9027); *Clostridium sporogenes* (CIP 7939); *Candida albicans* (ATCC10 231); and *Aspergillus brasiliensis* (ATCC16 404) (Eurofins). *Escherichia coli* (NCTC 13167) was added, as it is a common microorganism. Fifty colony-forming units (CFU) of each microorganism were added to 50 mL of TSB (Biomerieux). TSB that was not inoculated was used as a negative control. Additionally, an applicability test with the NE was carried out with the microorganisms by filtering 50 CFU of each microorganism.

#### 2.1.8. Endotoxin Test

Endotoxin determination was based on the kinetic chromogenic LAL test based on enzymatic substrate cleavage in the presence of endotoxins in conformity with one of the prescribed methods of the European Pharmacopeia [17]. The Endosafe^®^ portable test system (Nexgen-PTS™, Charles River, Massachusetts, USA) was used, and the endotoxin limit for propofol was set to no more than 0.33 endotoxin units/mg, in accordance with the USP monograph of propofol emulsions [18] and the K/M ratio provided by the European Pharmacopeia. The validation of the assay was based on the determination of the Maximum Valid Dilution (MVD) according to Equation (1), both provided by Ph.Eur.2.6.14 and USP 43-NF 38 <85 > [19], where λ is the sensitivity of the cartridge, which was claimed to be in the range 5.00–0.05 endotoxin units/mL by the provider (Limulus Amebocyte Lysate Endosafe^®^ PTS, Charles River, MA, USA).

Equation (1). Maximum Valid Dilution (MVD)
*MVD* = *(endotoxin limit × concentration of the solution)*/*λ*(1)

The MVD for propofol 20 mg/mL was calculated to be 132. The pretreatment of the propofol NE sample was determined to ensure the lack of interference with the chromogenic method. Two pretreatment conditions were assessed: simple water dilution at 1:125 (*v*/*v*) or breaking the NE with methanol followed by centrifugation (3500 rpm, 10 min) and the analysis of the supernatant. The control solution was pyrogen-free water for injection. All samples were pH-adjusted with NaOH (0.5 N) in the range of 6.5 to 7.8. The acceptance criteria for a valid chromogenic test were spike recovery in the range of 50–200% and a confidence interval < 25%.

## 3. Results

### 3.1. NE Process Selection

#### 3.1.1. Extemporaneous Addition

When propofol oil was mixed directly with the Intralipid^®^ NE, visual observation showed macroscopic oil droplets on the surface of the NE, as shown in Figure 1A. After a few days, phase separation was clearly seen (Figure 1B).

The granulometric characteristics of the extemporaneous addition of propofol to the Intralipid^®^ 20% NE are given in Table 3.

The mean diameters, PDI, zeta potential and pH of the Intralipid^®^ native NE were not affected by the addition of the propofol oil solution, but very large droplets were detected with the help of laser diffraction.

Macroscopic and granulometric analysis showed that the direct extemporaneous addition of propofol to a constituted injectable NE is not a suitable method for propofol NE preparation.

#### 3.1.2. “De Novo” Formulation

For the “de novo” formulation, the aim was to optimize both the mean droplet size and initial pH adjustment prior to heat sterilization.

Determination of the minimum number of passes through Microfluidizer^®.^

According to the propofol formulation, Figure 2 shows that seven passes is optimal with an mds around 160 nm. Additional passes did not add value to the final granulometric characteristics.

Determination of the optimal pH adjustment prior to heat sterilization of the propofol NE.

The macroscopic observation of the first panel of formulations quite clearly showed the impact of heat sterilization on the propofol NE when the initial pH (5.2) was not adjusted prior to heat sterilization, where the NE was broken with visual phase separation, and pH dramatically decreased to pH 4 after sterilization. The second round of tests focused on optimizing the adjusted pH between 7.5 and 9.5. Based on the results of the granulometry and the pH after sterilization, the optimal pH before sterilization was between 8.5 and 9.5 (Table 4).

Table 4 shows the direct impact of pH adjustment prior to sterilization on granulometric and macroscopic characteristics. Figure 3 gives an example of the macroscopic aspect of the 2% propofol nanoemulsion after heat sterilization with pH adjusted to 9.5 prior to sterilization. The optimal adjustment was set between 9.0 and 9.5 in our preparation condition, resulting in a final pH after heat sterilization between 6.0 and 6.5, which is suitable for clinical administration. Moreover, the pH specification in the USP monograph for a propofol injectable emulsion was revised from “between 7.0 and 8.5” to read “between 4.5 and 8.5”, which allows a quite large tolerance for the pH adjustment of the formulation.

### 3.2. Propofol HPLC-UV

#### 3.2.1. Validation of the Method

The method was validated according to ICH Q2 (R1) guidelines by evaluating, over 3 days, the linearity (correlation coefficient (r^2^)), the precision (coefficient of variation, CV%) and the accuracy (%). The average calibration curve over the 3 days of analytical validation was linear over the concentration range 12, 16, 20, 24 and 30 µg/mL, with a calculated correlation coefficient r^2^ of >0.99 (0.9996, 0.9994, and 0.9967 for day 1, day 2, and day 3, respectively). The intra- (Table 5) and inter-day (Table 6) variability, expressed as CV%, remained less than 2%, ranging from 0.41% to 0.69% and from 0.54% to 1.24%, respectively. Intra- and inter-day accuracies were consistent (between 98% and 102%, respectively), ranging from 98.06% to 101.07% and 99.74% to 100.13%, respectively.

Method validation statistics are given in Table 7. The homogeneity of variance was determined by the Debrown–Forsythe test and showed homogeneity of variance in the studied interval of the calibration curve with *p* > 0.05. For the calibration curve for day 1, the goodness of fit was found acceptable, and the CV% of the 3-day coefficient curves was satisfactory. The limit of detection (LOD) was determined from the background noise and was considered to correspond to the concentration at which the signal-to-noise ratio was 3:1. The limit of quantification (LOQ) was determined from a signal-to-noise ratio of 10:1.

#### 3.2.2. Stability-Indicating Method

Stress conditions leading to the maximum degradation of propofol were retained to ensure that our method was stability-indicating.

For the Pharma Chemicals reference (Figure 4A) used for the degradation study, only one peak at 3.8 min corresponding to propofol was observed.

For the Sigma reference (Figure 4B), three peaks were observed: a propofol peak at 3.8 min and two other peaks at 4.5 min and 8.5 min (degradation products and impurity, respectively).

For the basic degradation of the Pharma Chemicals reference with NaOH degradation (Figure 4C), three peaks are observed: a propofol peak at 3.8 min and two other degradation peaks at 1.9 and 4.5 min.

For oxidative degradation with H_2_O_2_ of the Pharma Chemicals reference (Figure 4D), four peaks are observed: a propofol peak at 3.8 min and three other degradation peaks at 4.2, 4.5 and 8.7 min.

Under acidic and photolytic stress conditions, no degradation products were observed [20].

Under basic and oxidative degradation conditions, the chromatographic peak area of propofol decreased by 25.7% and 29.4%, respectively, compared to the reference solution peak. The chromatographic peak purities of propofol (Pharma Chemical) after basic and oxidative degradation (peak match values of 957 and 959, respectively) are similar to the purity peak of the control (Pharma Chemical) before degradation (peak match value of 945), and all peak match values are close to the maximum peak match value (1000) when the peak start and peak end correspond to 100% of the spectrum in the peak maximum.

These results show that the predominant degradation pathway of propofol is hydrolysis in basic and oxidizing solutions. The validation of the different degradation conditions of propofol allows our analytical method to be used for a chemical stability study of propofol solutions in methanol, according to the ICH guidelines for stability-indicating methods.

### 3.3. Impact of Heat Sterilization on Propofol Stability

Propofol NEs properly pH-adjusted and heat-sterilized showed satisfactory results for both the physical granulometric properties of the NE and the chemical stability of propofol. The mean initial concentration, regardless of the pH, was 18.68 ± 0.36 µg/mL. Table 8 presents the comparative results of the relative percentage areas of the propofol (Sigma raw material) peak and degradation peak at 8.5 min before and after heat sterilization at 121 °C for 15 min. One can notice that the use of the Sigma raw material implies the initial presence of about 2% of the degradation product at 8.5 min. This peak was not present for the Pharma Chemical raw material. For a comparison of the effects of the process and stability, it is important to keep this initial difference in mind when interpreting the results.

Only slight degradation was noticed, and the maximum was found for the formulation pH-adjusted to 7.5. Degradation was under 3% for both pH 9.0 and 9.5 formulations.

### 3.4. Impact of EDTA in the Formulation on Propofol Stability after Heat Sterilization

The lack of EDTA in the formulation (performed with the Sigma raw material) when heat-sterilized induced a noticeable increase in degradation products at a retention time of 8.5 min, which was quantified at 20%. Comparative chromatograms of propofol from NE formulations with and without EDTA before and after heat sterilization are given in Figure 5.

The protective effect of EDTA on the appearance of degradation products when using our sterilization facility after heat sterilization was confirmed on commercial propofol emulsions. In Lipuro^®^ (formulated without EDTA), the same degradation peak at 8.5 min was found after heat sterilization, whereas in Diprivan^®^ (formulated with EDTA), no degradation peak appeared after heat sterilization (Figure 6).

### 3.5. Impact of Sterilization Processes on Physical and Chemical Stabilities

Heat sterilization: The acidification of the NE after heat sterilization was noticed, especially when the initial pH was not adjusted above 8.5 (Table 4). When the pH dropped to 4, both the growth of droplets and visual breakage were observed. The basic pH adjustment prior to sterilization allowed the limitation of the pH drop and the breakage of the NE. With regard to propofol’s chemical stability (Table 8), heat sterilization had no substantial effect on formulations where pH was adjusted above 8.5 prior to sterilization. When the pH was initially adjusted to a lower pH, i.e., 7.5, the percentage of the degradation peak was higher but remained within 5%.

Sterilization with 0.22 µm filter:

Filtration with a 0.22 µm filter as an alternative method for the sterilization of the NEs was suitable, as both the granulometric characteristics and final propofol concentration were preserved (Table 9).

### 3.6. Short-Term Propofol Chemical Stability

After one month, no evolution of the granulometric characteristics or chemical stability of propofol was noticed. In comparison to the D_0_ concentrations after heat sterilization, whatever the pH, the mean final propofol concentration was 18.77 ± 0.34 µg/mL. No evolution above 5% of the degradation peak was observed (Table 10). With regard to the pH, only a slight pH reduction of around 0.25 was found, but for all formulations, the pH stayed within the USP acceptance range of 4.5–8.5.

### 3.7. Free Propofol Determination

The total propofol concentration of the experimentally analyzed sample was 9.9 ± 0.045, 19.5 ± 0.087 and 20.7 ± 0.160 mg/mL (± SD; *n* = 2) for Diprivan^®^ 1%, Diprivan^®^ 2% and the in-house 2% propofol preparation, respectively. These concentrations were consistent with the expected final concentrations. The best final formulation, prepared by high-pressure homogenization and pH-adjusted prior to heat sterilization, showed a similar concentration of free propofol in the aqueous phase to the commercial propofol Diprivan^®^ 2% (Table 11).

The free propofol concentrations of the 2% preparation and 2% Diprivan^®^ are similar at 52.7 µg/mL and 53.4 µg/mL, respectively. The free propofol concentration of 1% Diprivan^®^ (28 µg/mL) is essentially proportional to the total propofol concentration. On the other hand, the proportion of free propofol is similar for the 2% propofol preparation, 2% Diprivan^®^ and 1% Diprivan^®^ at 0.26%, 0.27% and 0.28%, respectively.

In conclusion, the 2% propofol preparation has the same quantitative characteristics (total propofol concentration, free propofol concentration and proportion of free propofol/total propofol ratio) as 2% Diprivan^®^.

## 4. Discussion

### 4.1. Propofol Formulation Choice

A Diprivan^®^-like formulation was the most rational choice considering the emergency of the crisis and the need to provide a drug as close as possible to the marketed drug, considering the risk of major differences in pharmacokinetics and pharmacodynamics from non-emulsion formulations. For example, some works evaluated cyclodextrin [21] at 10 mg/mL as a potential alternative to non-emulsion formulations. However, in a review in 2005 [22], the design of non-emulsion formulations, i.e., a micellar solution, beta-cyclodextrin, soluble propofol prodrug, or microemulsion, exhibited variable outcomes, mainly increased pain. For example, Aquafol^®^, a microemulsion, showed an increased frequency and severity of pain on injection, which was associated with a higher aqueous-free propofol concentration of 63 µg/mL compared to 12 µg/mL in the LCT propofol emulsion [23]. On the other hand, some authors have developed a propofol microemulsion with macrogol(15)-hydroxystearate surfactant (Solutol^®^ HS15) as a substitute for lecithin [24] and were able to demonstrate in a clinical trial a moderate benefit for pain compared to a soybean NE. However, clinical evaluations of other formulations were limited, so we decided to stick to the ingredients of the reference Diprivan^®^ formulation, as it is known that differences in the oil components, or surfactants, may modify pharmacokinetics [25,26].

Taking into account that our objective was not long-term stability, we assessed the chemical stability for 1 month after heat sterilization for an initial pH adjusted to 7.5, 8.5, 9.0 and 9.5 and tested the propofol formulation without EDTA, which is used as a microbiological preservative [27]. However, we found some differences in propofol stability between formulations including EDTA and formulations without EDTA. A slight yellow color was visually observed in the formulation without EDTA compared to the formulation with EDTA, which was perfectly white. In addition, we noticed a degradation peak after autoclaving that was much lower when EDTA was present in the formulation. In addition to the preservative effect, an antioxidant effect can be expected [28] from EDTA. To limit drug oxidation, propofol NEs are usually manufactured under a nitrogen atmosphere. We considered the filling process under a nitrogen atmosphere to be an additional issue for hospital preparation, so the vials were filled under a normal atmosphere, which may favor the oxidation of propofol; in this context, EDTA seemed to be helpful and may be recommended for the hospital process. In addition, as was clearly demonstrated by Han and Washington [29], EDTA has no adverse effects on the physical stability of the emulsion.

The pH adjustment of the water phase of the emulsion prior to heat sterilization was a key factor to guarantee a pH in the range of 4.5–8.5, as recommended by USP. A pH drop due to heat sterilization is a well-known effect previously analyzed by Herman and Groves [30] and can be attributed to the partial hydrolysis of egg phospholipids, leading to the release of free fatty acids. Acidification may be amplified by oxygen if a nitrogen atmosphere is not used for the emulsion preparation. In our conditions, the adjustment of the pH between 9 and 9.5 prior to heat sterilization was optimal and gave a pH of around 6 after heat sterilization, which is in the range of the expected pH of the formulation, between 4.5 and 8.5. Moreover, we noticed that insufficient initial pH adjustment was associated with increased propofol degradation products, with the maximum at pH 7.5. This degradation may be related to oxidation being favored by the combination of the pH drop, fatty acid release and high-temperature exposure. Taken together, the physical and chemical stability results justified the choice of an initial pH adjustment above 9.0 prior to heat sterilization.

It is worth noting that if terminal filtration is used instead of heat sterilization, pH adjustment is not required since the initial pH of the formulation is around pH 6 and suitable for intravenous administration.

### 4.2. Process of Emulsification and Size Reduction

Unsurprisingly, the extemporaneous addition of the propofol liquid oil solution to a commercial Intralipid^®^ NE was not successful, showing macroscopic oil droplets and quick phase separation. This method, previously used for other drugs such as amphotericin B [31], must be considered a dangerous practice in the case of propofol. This difference may be explained by the chemical differences between the drugs: the former may bind to and be located at the phospholipid interface of the droplets [32], whereas propofol oil may not. Finally, with respect to the emulsification process, the de novo preparation of the propofol NE, combined with the proper reduction in droplet size, was the only method suitable to prepare the propofol NE for intravenous administration. Both the inactive ingredients and the homogenization process play an important role in the granulometric characteristics of NE. With regard to the oil composition, the oil phases of Diprivan^®^ and Propofol-Lipuro^®^, prepared with soybean oil and a mixture of 50:50 soybean/MCT, respectively, showed slight differences in granulometry. The mean droplet size, experimentally measured by DLS, for Diprivan^®^ 1% and 2% and Propofol-Lipuro 1% and 2% had mean diameters of 175; 168; 189; and 184 nm, respectively [33]. In addition, the maximum pressure of the homogenization device modified the granulometric results. An optimization study using a factorial design to optimize the pressure from 650 to 850 bar for a 1% propofol NE formulation similar to Diprivan^®^ 1% revealed the best result at the highest pressure with a 174 nm NE mean diameter and a PDI < 0.1, in conformity with commercialized Diprivan^®^ 1% [34]. The Microfluidizer^®^ allowed us to drive the homogenization at higher pressure (1500 bar), producing the overall lowest mean diameter of 150–160 nm in NEs.

Thanks to the granulometric results, it can be envisioned that hospital preparation can use aseptic preparation combined with final sterile filtration instead of heat sterilization. According to our granulometric results, the NE can be sterilized with a 0.22µm filter, as was previously demonstrated for parenteral emulsions [34] and, more recently, for a 1% propofol NE [35]. The best method to obtain a suitable droplet size was the use of a high-pressure homogenizer at 1500 bar. It resulted in a mean diameter of around 150–160 nm with a very narrow distribution of the droplets (pdi < 0.1) and the absence of droplets larger than 220 nm, which could rapidly clog the filter pores. Nevertheless, the aseptic process imposes more constraints related to the final sterilization process, with the need to work in conditions with the highest level of environmental control, grade A/B [36]. This can be balanced with the available hospital facilities, and the heat sterilizer may not be available in hospital pharmacies with low-scale production.

### 4.3. Safety Consideration with Regard to Free Propofol

Searching for free propofol in the aqueous phase was based on the knowledge of, firstly, excess phospholipids in emulsion formulations [37,38], which helps in the dispersion of traces of free propofol in water, as well as the knowledge of the concentration of free propofol of various commercial emulsions previously published. According to Muller RH et al. [11], for Diprivan^®^ 1%, the mean total propofol concentration determined was 9.84 mg/mL, and the mean free propofol concentration determined was 19.76 µg/mL. Our analysis of Diprivan^®^ 1% essentially confirmed the literature data.

Our results showed that the free propofol in our “de novo” formulation was very close to that in the commercial formulation of Diprivan^®^ 2%. Notably, this concentration was almost doubled compared to Diprivan^®^ 1% and looks proportional to the final propofol concentration. The amount of free propofol is approximately 0.25% of the total propofol concentration. Moreover, a recent structural study of four commercial propofol emulsions [39] showed a coexisting mixture of oil droplets, lipid vesicles and droplet lipid vesicle aggregates. However, propofol was demonstrated to be predominantly located in oil droplets, and the lipid vesicle fraction was negligible, which did not affect therapeutic equivalence.

We can safely conclude from our assay that the free propofol traces found in our “de novo” NE formulation will not modify the tolerance profile compared to commercial formulations.

### 4.4. Controls and Pharmaceutical Release

Undoubtedly, the quality of small batches produced in hospital pharmacies should be based on extensive quality controls, including sterility testing, endotoxin testing and final propofol concentration controls. All of these methods were validated in our study for the propofol NE. Additionally, particle counting and visual observation for visible particles must be performed. As it may be difficult to provide a dynamic light scattering apparatus, it may be suggested that 1.2 µm filtration be systematically added to the administration line.

### 4.5. Feasibility of Propofol Production within Hospital Facilities

Hospital pharmacies all over Europe are used to produce ready-to-administer (rtA) parenteral formulations on different scales, i.e., a small or large scale. Small-scale pharmacies usually focus on the preparation of rtA via an aseptic process using commercialized sterile drugs as raw materials. Large-scale-preparation pharmacies are intended for ready-to-use (RtU) preparations and are more likely to be able to provide sterile drugs from non-sterile raw materials by using heat sterilization.

The need for a high-pressure homogenizer dramatically limits the access of small-scale compounding pharmacies to propofol production. Considering large-scale-production hospital pharmacies, it may be useful to include it in the production equipment to allow the preparation of parenteral emulsions for small batches, allowing them to face shortages of parenteral or non-parenteral, i.e., ophthalmic, drugs using emulsion technology. It could also pave the way for the development of clinical trials in the future using nanoemulsion technology.

## 5. Conclusions

A proof of concept of propofol NE preparation in a hospital pharmacy was demonstrated only for the de novo emulsification process combined with a high-pressure homogenizer for the proper nanoscale droplet size reduction for IV administration. Sterilization by either heat or filtration can be envisioned, showing perfect physical characteristics of the nanoemulsion and the chemical stability of propofol. Both the physical characteristics and the very low free propofol concentration in the aqueous phase confirmed the suitability of the in-house formulation of the 2% propofol NE for the substitution of commercial formulations in the case of drug shortages. In addition, methods for testing the NE produced at the hospital scale were successfully developed and included sterility, endotoxin testing and the determination of the final propofol concentration. Hospital pharmacies can be a useful support in the case of an acute crisis leading to dramatic drug shortages from the pharmaceutical industry.

## Figures and Tables

**Figure 1 pharmaceutics-15-00905-f001:**
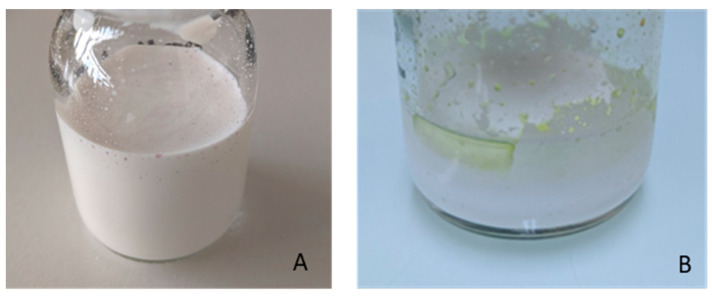
NE of propofol after extemporaneous addition to Intralipid^®^ 20%. (**A**) Immediately after the addition and (**B**) a few days later.

**Figure 2 pharmaceutics-15-00905-f002:**
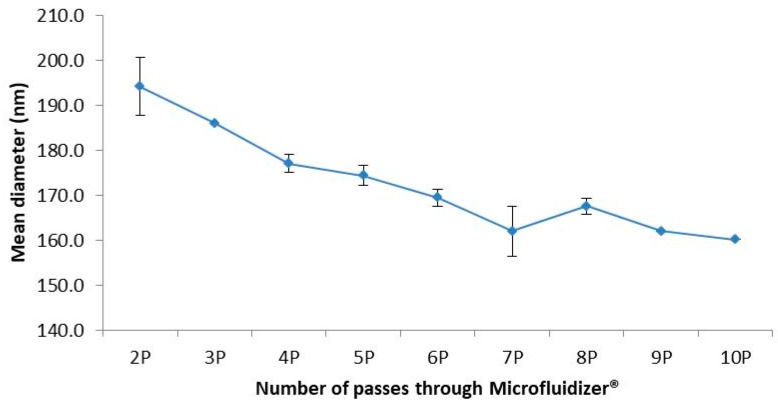
Optimization of the number of passes through Microfluidizer^®^.

**Figure 3 pharmaceutics-15-00905-f003:**
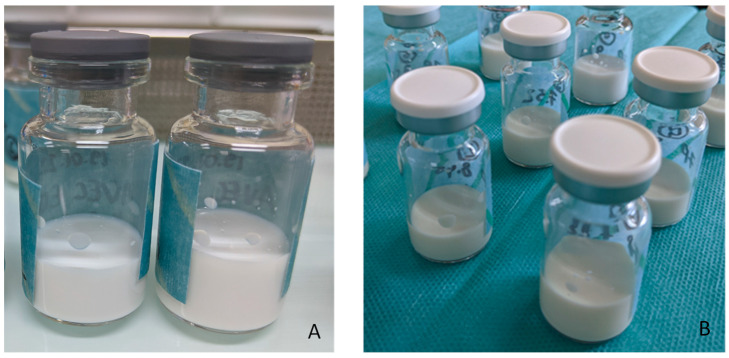
NE of propofol before (**A**) and after (**B**) heat sterilization (pH 9.5 adjusted prior to sterilization).

**Figure 4 pharmaceutics-15-00905-f004:**
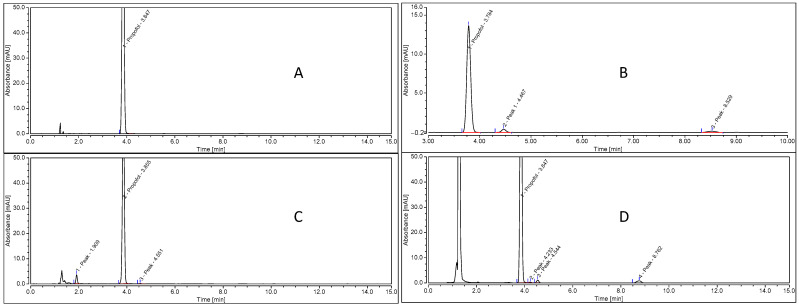
Chromatograms of propofol before degradation under stress conditions ((**A**), Pharma Chemicals reference; (**B**), Sigma reference) and after degradation under stress conditions ((**C**), Pharma Chemicals reference with NaOH degradation; (**D**), Pharma Chemicals reference with H_2_O_2_ degradation).

**Figure 5 pharmaceutics-15-00905-f005:**
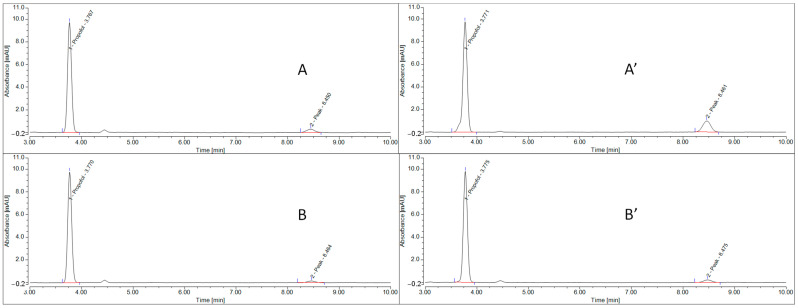
Propofol chromatograms of NE formulations without EDTA before (**A**) and after (**A’**) heat sterilization and with EDTA before (**B**) and after (**B’**) heat sterilization.

**Figure 6 pharmaceutics-15-00905-f006:**
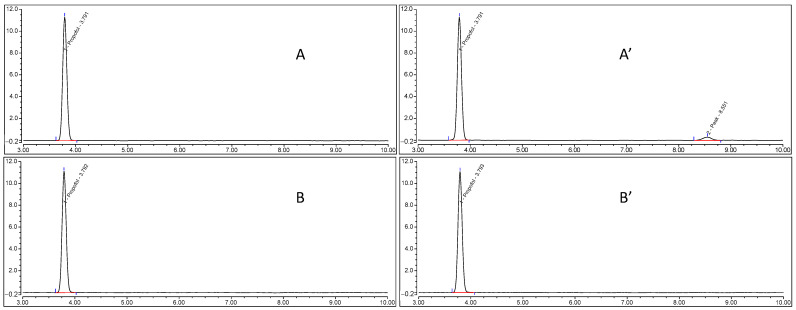
Chromatograms of Propofol-Lipuro^®^ before (**A**) and after (**A’**) heat sterilization and Diprivan^®^ before (**B**) and after (**B’**) heat sterilization.

**Table 1 pharmaceutics-15-00905-t001:** Composition of propofol NEs.

	Diprivan^®^	“>De Novo” Formulation	“Extemporaneous Addition”
Propofol	2 g	2 g	2 g
Intralipid^®^ 20%	-	-	qs 100 mL
Soybean oil	10 g	10 g	
Egg lecithin	1.2 g	1.2 g	
Glycerol	2.25 g	2.25 g	
Edetate disodium	0.0055 g	±0.0055 g	
Sodium hydroxide	qs to adjust pH	qs to adjust pH	
Water for injection	qs 100 mL	qs 100 mL	

**Table 2 pharmaceutics-15-00905-t002:** Time selection for dialysis.

Time (Minutes)	240	330	420	480
Concentration mg/mL	0.0487 ± 0.0011	0.0540 ± 0.0019	0.0556 ± 0.0005	0.0543 ± 0.0037

Results are mean ± SD of 6 experiments.

**Table 3 pharmaceutics-15-00905-t003:** Characteristics of extemporaneous addition of propofol in Intralipid^®^ 20% NE.

Raw Material	MDS (nm)	PDI	Droplets > 1 µm (%)	Droplets > 5 µm (%)	Zeta Potential (mV)	pH
Intralipid^®^	290.6 ± 0.8	0.143 ± 0.002	0	0	−44.5 ± 2.1	7.51 ± 0.02
Propofol ^1^	290.7 ± 1.1	0.160 ± 0.005	2	0	−43.1 ± 0.4	7.27 ± 0.11
Propofol ^2^	294.0 ± 1.2	0.131 ± 0.019	-	7	−42.2 ± 0.8	7.39 ± 0.05

^1^ Sigma; ^2^ Pharma Chemicals.

**Table 4 pharmaceutics-15-00905-t004:** Characteristics of NEs before and after autoclaving related to initial pH adjustment.

	Before Heat Sterilization	After Heat Sterilization
pH	5.20 (±0.04)	3.98 (±0.02)
mds	161.6 (±1.2)	816.9 (±67.1)
pdi	0.0787 (±0.013)	0.0987 (±0.0175)
pZ	−41.7 (±2.3)	−44.8 ((±0.5)
Visual observation	Homogeneous	Phase separation
pH	7.59 (±0.01)	4.74 (±0.05)
mds	153.4 (±2.4)	210.1 (±12.4)
pdi	0.060 (±0.010)	0.078 (±0.008)
pZ	−34.3 (±4.2)	−42.6 (±2.82)
Visual observation	Homogeneous	Homogeneous
pH	8.49 (±0.04)	5.59 (±0.05)
mds	152.2 (±0.5)	165.9 (±4.2)
pdi	0.061 (±0.003)	0.070 (±0.002)
pZ	−33.7 (±3.9)	−41.2 (±2.3)
Visual observation	Homogeneous	Homogeneous
pH	8.97 (±0.01)	5.95 (±0.05)
Mds (nm)	150.9 (±0.8)	155.3(±2.4)
pdi	0.055 (±0.004)	0.057 (±0.012)
pZ (mV)	−34.9(±2.2)	−39.7 (±2.4)
Visual observation	Homogeneous	Homogeneous
pH	9.47 (±0.01)	6.62 (±0.07)
Mds (nm)	149.8(±1.8)	152.4 (±1.2)
pdi	0.055 (±0.009)	0.064 (±0.007)
pZ (mV)	−33.3 (±4.4)	−39.0 (±1.0)
Visual observation	Homogeneous	Homogeneous

**Table 5 pharmaceutics-15-00905-t005:** Results of intra-day analyses of 3 quality control samples of various concentrations of propofol solutions in methanol.

Day	Theoretical Concentration µg/mL	Observed Concentration µg/mL Mean ± SD	Accuracy %	Precision CV%	n
1	14	14.40 ± 0.08	99.30	0.57	6
	22	21.90 ± 0.11	100.89	0.48	6
	28	29.80 ± 0.17	99.7	0.58	6
2	14	14.40 ± 0.07	99.41	0.47	6
	22	22.20 ± 0.09	99.43	0.41	6
	28	28.80 ± 0.18	101.07	0.63	6
3	14	13.90 ± 0.10	99.53	0.69	6
	22	21.90 ± 0.12	98.06	0.55	6
	28	29.90 ± 0.15	100.28	0.49	6

**Table 6 pharmaceutics-15-00905-t006:** Results of inter-day analyses of 3 quality control samples of various concentrations of propofol solutions in methanol.

Over 3 Days	Theoretical Concentration %	Observed Concentration Mean % ±SD	Accuracy %	Precision CV%	n
QC1	100	99.41 ± 0.54	99.74	0.54	18
QC2	100	99.46 ± 1.24	99.97	1.24	18
QC3	100	99.41 ± 0.76	100.13	0.76	18

**Table 7 pharmaceutics-15-00905-t007:** Method validation statistics.

Homogeneity test of variance	*p* = 0.3548
Goodness of fit	R-Squared 0.9996, root-mean-square error 7.933 × 10^−2^
CV% coefficient curves (3 days)	<0.2%
LOD	0.00114 mg/mL
LOQ	0.00381 mg/mL

**Table 8 pharmaceutics-15-00905-t008:** Effect of heat sterilization and pH adjustment on chemical stability of propofol *.

	Propofol Peak (%)		Degradation Peak at 8.5 min (%)		Degradation Peak Difference (%) before/after Heat Sterilization
	Before heat sterilization	After heat sterilization	Before heat sterilization	After heat sterilization	
pH 7.5	97.80 ± 0.37	93.73 ±0.66	2.20 ± 0.37	6.27 ± 0.66	+4.07
pH 8.5	97.97 ± 0.29	94.88 ± 0.66	2.03 ± 0.29	5.12 ± 0.44	+3.09
pH 9.0	97.90 ± 0.26	95.17± 0.59	2.10 ± 0.26	4.83 ± 0.59	+2.73
pH 9.5	97.93 ± 0.29	95.40 ± 0.29	2.07 ± 0.35	4.60 ± 0.29	+2.53

Results are mean ± SD of 3 NE samples; each sample measurement was repeated 6 times. * Sigma propofol raw material.

**Table 9 pharmaceutics-15-00905-t009:** Granulometric characteristics and propofol concentration before and after 0.22µm filtration.

	Before 0.22 µm Filtration	After 0.22 µm Filtration
Mean diameter (nm)	159.3 ± 0.3	164.2 ± 0.5
PDI	0.061 ± 0.015	0.064 ± 0.017
Propofol concentration (µg/mL)	20.90 ± 0.73	20.47 ± 0.82

Results are mean ± SD of 3 samples.

**Table 10 pharmaceutics-15-00905-t010:** Chemical stability of propofol heat-sterilized after 30 days of storage at room temperature.

	Propofol Peak (%)	Degradation Peak at 8.5 min (%)	Degradation Peak Difference (%) D_0_/D_30_
pH 7.5	94.86 ± 2.40	5.14 ± 2.40	−2.82
pH 8.5	95.72 ± 2.40	4.28 ± 1.53	−0.58
pH 9.0	95.83 ± 1.44	4.17 ± 1.44	−0.66
pH 9.5	95.86 ± 1.67	4.14 ± 1.67	−0.91

Results are mean ± SD of 3 NE samples; each sample measurement was repeated 6 times.

**Table 11 pharmaceutics-15-00905-t011:** Free propofol in the in-house prepared NE in comparison with commercialized Diprivan^®^ NEs.

	Free Propofol Concentration (mg/mL)	Free Propofol (%)
2% Propofol preparation	0.0527 ± 0.0014	0.26
Diprivan^®^ 2%	0.0534 ± 0.0004	0.27
Diprivan^®^ 1%	0.0280 ± 0.0004	0.28

Results are mean values of 6 determinations ± SD.

## Data Availability

Not applicable.

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
