# Peer review of "Hospital Production of Sterile 2% Propofol Nanoemulsion: Proof of Concept"

_pharmaceutics, 2023, doi:10.3390/pharmaceutics15030905_

Round 1
Reviewer 1 Report
Comments to authors,
1. In contrast with injectable solutions, NE are thermodynamically instable systems. Replace the word instable systems to stable systems.
2. Authors need to cite their abstract in the conference GERPAC 2022 conference, Hospital production of propofol injectable nanoemulsion: proof-of-concept
3. Other studies discussed propofol nanemulsions such as:
Comparative evaluation of propofol in nanoemulsion with solutol and soy lecithin for general anesthesia
Formulation and Evaluation of Multidose Propofol Nanoemulsion Using Statistically Designed Experiments
4. Discuss the difference between these studies and your study and add this discussion to the draft
5. Why the pH of the nanoemulsion decreased after sterilization, please discuss
6. Literature reported the nanoemulsion droplet size between 10-100 nm. Here the size is above 100 nm, please explain.
7. Propofol Emulsions properly pH-adjusted and heat sterilized showed satisfactory results for both physical granulometric properties of the emulsion and propofol chemical stability. Why authors are referring here to emulsions and not nanoemulsions? Also Figure 6, Table 10?
8. Why authors chose Sigma raw material for studying the impact of heat sterilization on Propofol stability (Section 3.3)
9. Why maximum degradation was found for the formulation 7.5 pH adjusted? Discuss and support your data.
10. Are the high pH values (7.5, 9.5, 9.0, and 9.5) suitable for intravenous administration?
11. Any idea about the size of Lipuro® and Diprivan® emulsions. Is it appropriate to compare those with your nanoemulsion formulation?
12. The total propofol concentration on the sample experimentally analyzed was 9.9 ± 0.045; 19,5 ± 0.087 and 20,7 ± 0.160 mg/mL, correct to 19.5 ± 0.087 and 20.7 ± 0.160.
13. In conclusion, the 2% propofol preparation has the same quantitative characteristics (total propofol concentration, free propofol concentration and proportion of free propofol/total propofol) than 2% Diprivan®. Why did you use “than” here? Is it the same, higher, or lower?
Author Response
The authors thank the reviewer for his fruitfull comments. Please see attachment to read point-by-point answers.
Your sincerely

Reviewer 2 Report
This is an intering manuscript focused on the validation of a method for the elaboration of propofol nanoemulsions. In general, the manuscript is well written and clear. It is suitable for publication in Pharmaceutics. Here there are some aspects comments that should be revised:
Major points:
1. I strongly suggest changing the tittle. I agree that it is interesting to include the information/utility of the elaboration method of this NE in drug shotage as occurred in COVID19 crisis. However, COVID19 is "a lateral aspect/additional discussion". The title should refect the objective/main part of the work that is the validation/development of propofol NE ..
2. The same comment for introduction. The authors should not start with COVID 19. But include this information latter as additional comment.
3. The authors can include validation satatistics ( at least linearity, lack of fit...) I suggest a table.
4. In table 7 I understand that the author include the difference in the % of the peaks of degradation product, right? This should be makerd.
5. Why did not the authors evaluate the effect of heat sterilization on physical stability?
Minor points:
- Figure 1 and 2 should be merged. They do not provide much information.
- Figure 4: The authors should include an image before and after.
Author Response

(The authors gave the same response as above.)

Round 2
Reviewer 1 Report
None.
Author Response
Dear reviewer, thank you again for your review. We have checked and corrected the English spelling throughout the manuscript. It is highlighted in yellow in our second revision.
Sincerely yours
Reviewer 2 Report
The authors have adressed all my previous comments and suggestions and have improved the quality of the manuscript. It deserves to be published. Just one comment.
Please, revise Figure 4. In the compiled file the cromatograms are too small. I can imagine that this is an error when pasting the figure to the manuscript file.
Author Response
Dear reviewer, thank you again for your review. We have checked and corrected the English spelling throughout the manuscript. It is highlighted in yellow in our second revision. In addition, we enlarged the chromatograms of the fig 4,5 and 6.
Sincerely yours